

# A simplified approach for efficiency analysis of machine learning algorithms

Muthuramalingam Sivakumar[1], Sudhaman Parthasarathy[2] and Thiyagarajan Padmapriya[2]

[1] Department of Computer Science and Engineering, Thiagarajar College of Engineering, Madurai, Tamil Nadu, India
[2] Department of Applied Mathematics and Computational Science, Thiagarajar College of Engineering, Madurai, Tamil Nadu, India

## ABSTRACT

The efficiency of machine learning (ML) algorithms plays a critical role in their deployment across various applications, particularly those with resource constraints or real-time requirements. This article presents a comprehensive framework for evaluating ML algorithm efficiency by incorporating metrics, such as training time, prediction time, memory usage, and computational resource utilization. The proposed methodology involves a multistep process: collecting raw metrics, normalizing them, applying the Analytic Hierarchy Process (AHP) to determine weights, and computing a composite efficiency score. We applied this framework to two distinct datasets: medical image data and agricultural crop prediction data. The results demonstrate that our approach effectively differentiates algorithm performance based on the specific demands of each application. For medical image analysis, the framework highlights strengths in robustness and adaptability, whereas for agricultural crop prediction, it emphasizes scalability and resource management. This study provides valuable insights into optimizing ML algorithms, and offers a versatile tool for practitioners to assess and enhance algorithmic efficiency across diverse domains.

## INTRODUCTION

Machine learning (ML) is a type of artificial intelligence in which computers learn from data instead of requiring explicit programming. The rapid advancement of machine learning has revolutionized numerous fields, from healthcare and finance to autonomous systems and natural language processing (*Kufel et al., 2023*). As the scope and complexity of ML applications increase, there is a need for efficient and effective algorithms. The efficiency of an ML algorithm, encompassing both its computational performance and predictive accuracy, is critical for its practical deployment. However, assessing and comparing the efficiency of these algorithms can be challenging because of the diverse nature of the tasks, datasets, and performance metrics involved (*Deng et al., 2021*; *Gao & Guan, 2023*).

Algorithms are fundamental instructions that guide computers in solving problems, acting as detailed steps for computation. They outline the precise sequence of steps

Corresponding author
Muthuramalingam Sivakumar, siv-siva.kumar.21@gmail.com

required to achieve the desired outcome (*Zhou et al., 2017*). Efficiency, a central concept in algorithms, focuses on using minimal resources (time and memory) to obtain the job. This translates into faster programs that are capable of handling ever-growing datasets. Time complexity, a measure of how long an algorithm takes to run, is crucial for tasks such as searching for massive libraries. Similarly, space complexity is related to memory usage. Efficient algorithms are the source of smooth-running software. They ensure faster programs that can efficiently handle large datasets efficiently (*Mehta et al., 2023*). This efficiency becomes even more important as the volume of data constantly increases. However, selecting the best algorithm involves a balancing act. A slightly less efficient method may be sufficient for smaller datasets. However, for larger datasets, the efficiency of ML algorithms is key. Understanding the trade-offs between different algorithms and their efficiencies is essential for designing effective and scalable software solutions (*Abbad Ur Rehman et al., 2021*; *Ali et al., 2023*).

Efficiency in terms of machine learning algorithms can be defined as a measure of an algorithm's performance across various dimensions, including computational resource usage, execution time, predictive performance, and adaptability to different data conditions (*Ali et al., 2021*). This multifaceted approach ensures that the algorithm is not only accurate but also practical for real-world application. Computational efficiency refers to the time and space complexities of an algorithm. Time complexity is the amount of time required by the algorithm to train a given dataset, which is often measured in the Big-O notation. Space complexity, on the other hand, is the amount of memory required by the algorithm during both training and inference phases. Efficient algorithms should minimize both time and space complexities, allowing for faster processing and reduced resource consumption (*Majeed, 2019*; *Faruque & Sarker, 2019*).

The predictive performance is another critical aspect that encompasses several metrics. Accuracy, which is the proportion of correct predictions made by the algorithm, is a fundamental measure. Precision and recall provide insights into the algorithm's performance on imbalanced datasets, with precision measuring the correctness of positive predictions, and recall measuring the ability to identify all positive instances. The F1-score, which is the harmonic mean of precision and recall, offers a single metric of effectiveness (*Saranya et al., 2020*; *Singh & Goyal, 2020*). Scalability denotes the ability of an algorithm to handle increasing amounts of data while maintaining its performance (*Singh & Goyal, 2020*). As datasets grow, scalable algorithms can process larger volumes without significant degradation in speed or accuracy, making them suitable for big-data applications (*Zhou et al., 2017*). Robustness is the capability of an algorithm to maintain its performance despite variations in the data, such as noise, outliers, or missing values. A robust algorithm can handle real-world data imperfections effectively, ensuring reliable performance under diverse conditions (*Choudhury & Bhowal, 2015*).

Generalization is the ability of an algorithm to perform well on unseen data, indicating its capacity to generalize from a training set to real-world applications. Algorithms that are well-generalized may avoid overfitting and are better suited for practical use, where data encountered during deployment may differ from the training data (*Gzar, Mahmood & Abbas, 2022*). Resource utilization involves the amount of computational resources

required during the training and prediction phases (*e.g.*, CPU and GPU). Efficient algorithms make optimal use of available hardware resources, reduce costs, and enable deployment in resource-constrained environments (*Ali et al., 2023*; *Dagli et al., 2022*). Ease of implementation and use reflects the complexity of the algorithm in terms of coding and deployment. Algorithms that are simpler to implement and require fewer hyperparameters or preprocessing steps are often preferred in practice (*Vazquez, 2022*). Adaptability refers to the flexibility of an algorithm to be fine-tuned or adapted for different types of data or tasks. Algorithms that can be easily customized or extended are more versatile and useful across various applications, making them valuable tools for practitioners (*Vakili, Ghamsari & Rezaei, 2020*; *Fernandez et al., 2018*).

By encompassing these dimensions, for machine learning algorithms, the term "efficiency" provides a holistic view of an algorithm's performance, highlighting not only its predictive accuracy, but also its practicality, scalability, and robustness (*Punia et al., 2021*). This comprehensive definition helps practitioners choose algorithms that are not only theoretically effective but also viable in real-world applications (*Bashir et al., 2021*). In this study, we propose a simplified approach to perform an efficiency analysis of machine-learning algorithms. Our method aimed to streamline the evaluation process by providing a standardized conceptual framework that can be universally applied across various ML tasks. This approach not only simplifies the comparison of algorithm performance, but also enhances the reproducibility of results, which is often a significant hurdle in the field of machine learning (*Ali et al., 2021*).

In machine learning, scalability refers to an algorithm's ability to maintain performance as the size of the dataset or the complexity of the task increases, making it essential for handling large-scale data efficiently. Robustness denotes the algorithm's capacity to perform reliably despite noise, outliers, or incomplete data, ensuring stability in real-world conditions, where data are often imperfect. Generalization is the ability of a model to apply what it has learned from the training data to unseen data, avoiding overfitting and ensuring that the model can handle new, dynamic environments. Finally, resource utilization involves the efficiency of an algorithm using computational resources, such as CPU, GPU, memory, and energy, which become critical in environments with limited hardware or power constraints, such as mobile devices or edge-computing platforms. Together, these factors contribute to the overall efficiency and practical applicability of machine learning algorithms.

The proposed conceptual framework addresses key aspects of algorithm efficiency, including training time, prediction speed, memory usage, and accuracy. By integrating these metrics into a cohesive analysis model, researchers and practitioners can make informed decisions regarding algorithm selection and optimization. This approach also facilitates the identification of trade-offs between different performance aspects and offers insights into how algorithmic improvements can be targeted (*Gzar, Mahmood & Abbas, 2022*). Through a series of experiments and case studies, we demonstrated the applicability and benefits of our simplified approach. By applying it to a diverse set of machine learning algorithms and datasets, we demonstrate its robustness and versatility (*Majeed, 2019*; *Faruque & Sarker, 2019*). This work not only contributes to the methodological toolkit of machine learning

researchers but also aims to bridge the gap between theoretical advancements and practical implementation (*Prager, 2023*). This study introduces a comprehensive yet simplified methodology for analyzing the efficiency of machine learning algorithms. By standardizing the evaluation process, we provide a valuable resource for machine learning research, promoting more efficient and effective algorithm development and deployment (*Ali et al., 2021*; *Bashir et al., 2021*).

## Motivation

The proliferation of ML algorithms across diverse domains such as healthcare and finance is driven by their potential to extract valuable insights from vast datasets. However, selecting an appropriate algorithm requires a thorough understanding of its efficiency and performance under specific conditions (*Faruque & Sarker, 2019*). Despite the availability of numerous algorithms, evaluating their efficiency remains complex owing to intricate statistical and computational processes. Traditional methods often focus on limited performance metrics, such as accuracy or F1-score, neglecting other crucial factors, such as computational cost and scalability (*Choudhury & Bhowal, 2015*; *Bashir et al., 2021*). This can lead to suboptimal choices, particularly for practitioners without deep ML expertise. This study addresses this gap by proposing a simplified, robust framework for evaluating the efficiency of ML algorithms. Our approach aimed to democratize the evaluation process, making it more accessible and practical for a broader audience. By considering a comprehensive set of performance indicators, we seek to improve decision making in algorithm selection and enhance the practical application of ML technologies.

## RELATED WORKS

Although time complexity is a valuable metric for analyzing the efficiency of machine learning algorithms, it has several drawbacks that limit its effectiveness as a standalone measure (*Vakili, Ghamsari & Rezaei, 2020*). One significant issue is that time complexity oversimplifies the analysis by providing an asymptotic upper bound on the running time relative to the input size, often expressed in the Big-O notation. This approach focuses on the worst-case scenario, ignoring constants and lower-order terms. As a result, it may not accurately reflect the actual running time for practical, finite datasets, potentially leading to misleading conclusions regarding an algorithm's efficiency. Another drawback of time complexity is the lack of practical context. Time complexity does not account for variations in the hardware, implementation details, or system architecture. The two algorithms with the same theoretical time complexity could have significantly different actual running times because of these factors. For instance, one algorithm might be more optimized for parallel processing or specific hardware accelerations, such as GPUs, resulting in better performance despite identical time complexities on paper.

Time complexity also fails to consider the characteristics of data, such as distribution, sparsity, dimensionality, and specific structures (*Gzar, Mahmood & Abbas, 2022*). An algorithm might have a promising time complexity in theory but performs poorly with real-world data that exhibit properties not accounted for in the theoretical analysis. This oversight can lead to suboptimal algorithm choices when handling practical datasets. Time

complexity typically focuses on the training phase of an algorithm, ignoring significant preprocessing steps and hyperparameter tuning, which can be time consuming. These additional steps are crucial in the machine-learning workflow and can substantially impact the overall efficiency of an algorithm (*Ali et al., 2023*). By neglecting them, the time complexity provides an incomplete picture of an algorithm's practical efficiency in the context of machine learning (*Ali et al., 2023*).

Time complexity also does not address how well an algorithm scales with increasing data sizes in real-world scenarios. It fails to consider the required computational resources, such as memory and parallel processing capabilities. Practical considerations, such as data loading time, I/O operations, and other factors, are not captured by time complexity, highlighting the need for empirical evaluations to understand the true efficiency of an algorithm. Finally, the actual implementation of the algorithm can significantly affect its performance. An algorithm with optimal time complexity may have poor implementation, resulting in suboptimal performance. Conversely, a well-implemented algorithm with a slightly worse theoretical time complexity might perform better in practice. This disparity underscores the importance of considering implementation quality in conjunction with the theoretical analysis. Thus, it is inferred that, while time complexity is a useful theoretical tool for understanding the efficiency of machine learning algorithms, it has significant limitations when applied to real-world scenarios (*Majeed, 2019*). This oversimplifies the analysis, ignores practical considerations, and does not capture the full picture of an algorithm's performance and efficiency. Therefore, it should be used in conjunction with other metrics and empirical evaluations to obtain a comprehensive understanding of the efficiency of an algorithm.

The field of machine-learning efficiency is constantly evolving, with various approaches emerging to address the challenges of resource optimization and performance. Below, we discuss the key areas of research and tools that complement the proposed conceptual framework for analyzing the efficiency of machine learning algorithms. Automated machine learning (AutoML) offers valuable techniques for streamlining efficiency in machine learning workflows (*Prager, 2023*). AutoML leverages meta-learning to analyze past performance data on various algorithms and datasets, thereby recommending the most suitable algorithm for a new task. This process significantly reduces the need for manual exploration and experimentation, thereby saving time and resources. In addition, AutoML can automate hyperparameter tuning by exploring different configurations and evaluating their performance, thereby identifying the optimal configuration that enhances a chosen metric (*e.g.*, accuracy or training time) for a specific algorithm and dataset (*Vazquez, 2022*).

However, AutoML is not the only tool available for optimizing machine-learning workflows. Platforms such as Amazon SageMaker Canvas and Google Cloud AutoML offer similar functionalities, enabling users to build and deploy models without requiring extensive machine-learning expertise. These services also provide capabilities for data preprocessing, feature engineering, and model selection, which are crucial for improving the overall efficiency and performance of machine-learning algorithms (*Ali et al., 2023*). Moreover, traditional machine learning tools such as Scikit-learn and TensorFlow continue to play a significant role in research and industry by offering a flexible and comprehensive

environment for custom model development. Although these tools require more manual intervention than AutoML platforms, they provide greater control and customization, which can be beneficial in scenarios requiring specialized model architectures or unique data processing pipelines (*Bashir et al., 2021*).

A comprehensive review of the existing literature on machine learning efficiency reveals a diverse set of methodologies and tools, each with its strengths and limitations. For example, while AutoML platforms excel in automating the model selection and hyperparameter tuning processes, they may not always match the performance of handcrafted models designed for specific tasks. Furthermore, reliance on predefined pipelines in these platforms may limit their applicability to highly specialized domains (*Zhou et al., 2017*). On the other hand, tools such as Amazon SageMaker Canvas provide a more integrated approach that combines data preprocessing, model building, and deployment in a single platform; however, they also require consideration of the trade-offs between ease of use and the depth of customization available (*Mehta et al., 2023*).

Our proposed conceptual framework seeks to bridge these gaps by offering a standardized approach to efficiency analysis that can be applied across various tools and methodologies, ensuring that the strengths of different approaches are leveraged while addressing their limitations. The efficiency of machine learning algorithms goes beyond accuracy; they also consider the resources required for training and deployment. Cost-sensitive learning addresses this challenge by incorporating costs associated with different types of errors or resource usage into the learning process. By assigning costs to mistakes and resource consumption, the algorithm can be optimized to minimize the overall cost while achieving the desired level of performance (*Fernandez et al., 2018*). Green machine learning is an emerging field that focuses on the development of energy-efficient algorithms. This area explores techniques such as model compression (reducing model size) and hardware-aware algorithm design to minimize the computational resources required for training and deployment (*Mehta et al., 2023*). This is particularly relevant in large-scale or resource-constrained environments.

Although accuracy is a crucial metric, efficiency in machine learning encompasses other factors as well. Latency-aware learning addresses the critical aspect of the prediction speed in real-time applications. This research area focuses on developing algorithms that can achieve good accuracy while ensuring a low latency (response time). Techniques, such as model pruning or quantization, can be employed to reduce the computational complexity of the model and improve the prediction speed (*Dagli et al., 2022*).

Interpretability and explainability are gaining importance in the realm of efficiency. Understanding how a model arrives at its predictions can help to identify potential biases or inefficiencies within the algorithm. This allows for targeted improvements that can enhance efficiency and overall model performance (*Gao & Guan, 2023*). By considering these factors alongside accuracy, practitioners can achieve a more holistic view of machine-learning efficiency. These areas of research highlight the multifaceted nature of machine learning efficiency. The proposed conceptual framework, by focusing on identifying key efficiency metrics and enabling real-time decision-making, complements existing

approaches and empowers practitioners to optimize their machine learning workflows effectively.

In the literature, the efficiency of ML algorithms has traditionally been evaluated through a trial-and-error process, lacking a standardized method for comprehensive assessment. However, the authors of this article introduced a novel approach by proposing a new formula for calculating the algorithmic efficiency. Unlike conventional trial-and-error methods, this formula provides a systematic conceptual framework that considers a wide range of metrics and factors, allowing for a more holistic evaluation of ML algorithms. By offering a structured and standardized approach to efficiency assessment, this formula enhances the reliability of evaluations, enabling researchers and practitioners to make informed decisions in algorithm selection, optimization, and deployment. This represents a significant advancement in the field, offering a more robust and objective means of gauging the efficiency of ML algorithms across various applications.

Although *Mishra (2023)* explored the use of parallel computing techniques to accelerate machine learning algorithms on big data, this approach requires complex parallel computing conceptual frameworks that may not be accessible or straightforward to implement for all users. This method emphasizes performance enhancement through distributed computing, which, while effective, contrasts with a simplified approach aimed at a broader applicability. Similarly, *Yogi et al. (2024)* focused on the scalability and performance of machine-learning techniques in the context of high-volume social media data analysis. Their work addressed specific challenges in handling large-scale data, but it did so through advanced techniques that may not be generalizable to a simplified efficiency analysis conceptual framework.

*Salman et al. (2023)* delved into parallel machine learning algorithms to enhance the processing of large datasets. However, their focus on parallel processing introduces a level of complexity that can be challenging to apply to simpler scenarios. *Padmanaban (2024)* also investigated scaling issues, particularly in large-scale data infrastructures, requiring intricate strategies that are beyond the scope of a simplified approach to efficiency analysis.

*Ogbogu et al. (2023)* examined energy-efficient machine learning acceleration by incorporating both hardware and software optimizations. While their work is valuable in optimizing machine learning processes, it involves sophisticated technology and design considerations that differ significantly from straightforward efficiency analysis. *Bhagat, Mishra & Dutta (2023)* provide an in-depth analysis of deep distributed computing, highlighting its complexities, which, like other studies, are not aligned with the goal of simplifying the efficiency analysis.

*Althati, Tomar & Malaiyappan (2024)* discussed scalable machine-learning solutions for heterogeneous data in distributed platforms by employing advanced techniques to manage complex, diverse datasets. This work, which is crucial for handling large-scale data environments, requires a level of sophistication that is not necessary for a simplified efficiency analysis. Finally, *Ghimire & Amsaad (2024)* presented a parallel approach to enhance the performance of supervised machine learning in multicore environments by introducing technical complexity that adds layers beyond what is needed for a simplified analysis.

In contrast to these studies, the proposed work focuses on providing a more accessible and straightforward method for efficiency analysis of machine learning algorithms. By avoiding the need for advanced infrastructure, parallel processing, or intricate algorithm modifications, this approach lowers the technical barrier, making it broadly applicable and easier to implement across various machine-learning scenarios. This distinction allows for a more practical and user-friendly analysis, particularly in resource-constrained or less-technical environments.

## RESEARCH OBJECTIVE

This study aims to develop and validate a simplified yet robust conceptual framework for analyzing the efficiency of ML algorithms. The specific objectives are:

- **Identify key efficiency metrics:** Determine the critical metrics for assessing ML algorithm efficiency.
- **Propose a simplified evaluation process:** Develop an accessible method for comparing different ML algorithms.
- **Conduct empirical studies:** Validate the proposed framework through experiments and case studies.
- **Facilitate enhanced decision-making:** Provide a tool for practitioners to make informed choices when selecting ML algorithms for deployment.

The objective of this study is to develop and validate a simplified yet robust conceptual framework for the efficiency analysis of machine learning algorithms. This conceptual framework strives to democratize the evaluation process by making it accessible and understandable to a broader audience, including practitioners with limited expertise in machine learning. With the help of the proposed conceptual framework, this research seeks to achieve the following specific objectives: identifying key efficiency metrics for assessing the efficiency of ML algorithms, suggesting a simplified evaluation process to compare different ML algorithms, conducting empirical studies to validate the proposed conceptual framework, and providing a channel for enhanced decision-making by practitioners while choosing an ML algorithm for deployment. By achieving these objectives, this study aims to bridge the gap between theoretical analysis and practical application, facilitating the adoption of efficient machine learning algorithms across various domains and ensuring their effective use in solving real-world problems.

## PROPOSED CONCEPTUAL FRAMEWORK FOR EFFICIENCY ANALYSIS OF MACHINE LEARNING ALGORITHMS

The conceptual framework proposed for the efficiency analysis of machine learning algorithms in this study is based on three types of sources:

(1) Relevance of time complexity for assessing the computational efficiency of the ML algorithm (*Majeed, 2019*)

(2) Previous studies on attributes that contribute to the efficiency of ML algorithms (*Deng et al., 2021*) and

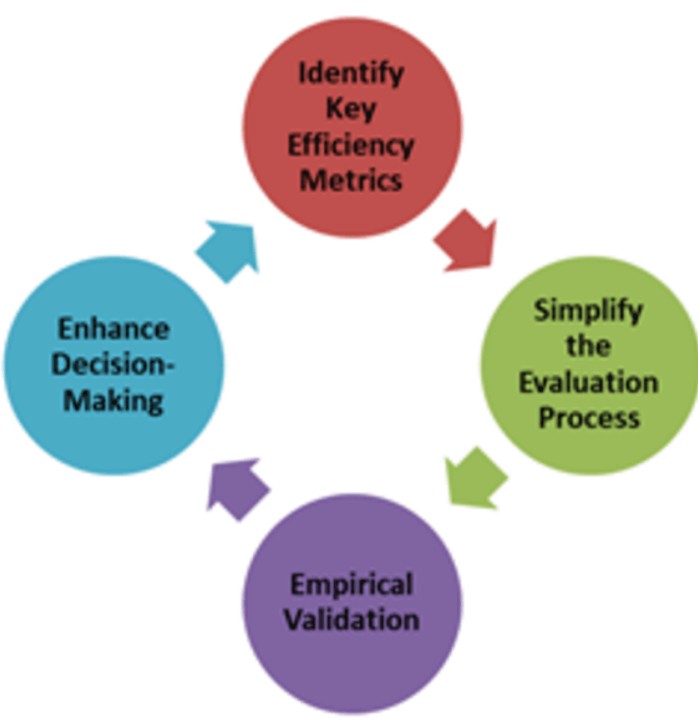

**Figure 1** **Conceptual framework for efficiency analysis of machine learning (ML) algorithms.** Conceptual framework for analysing machine learning algorithms.

(3) The need for a comprehensive evaluation of ML algorithms, simplification of accessibility, empirical validation, actionable insights for decision-making, and iterative refinement (*Gzar, Mahmood & Abbas, 2022*).

Figure 1 shows the conceptual framework for the efficiency analysis of ML algorithms.

The proposed conceptual framework (Fig. 1) has four components for analyzing the efficiency of machine learning algorithms. The first step is to identify key efficiency metrics to establish a comprehensive set of performance indicators that go beyond traditional metrics such as accuracy and F1-score. This includes evaluating the computational efficiency (time and space complexity), scalability, robustness, resource utilization, ease of implementation, and adaptability. By identifying these key metrics, the conceptual framework provides a holistic view of an algorithm's performance, ensuring that all critical aspects are considered in the evaluation process. This comprehensive set of metrics enables practitioners to make well-rounded assessments of machine-learning algorithms. The next step is to create straightforward methodologies and tools that allow users to evaluate and compare the efficiency of different machine-learning algorithms without requiring extensive technical knowledge or complex statistical analysis. This involves the development of user-friendly interfaces, automated tools, and clear guidelines that simplify the evaluation process.

The third step in the conceptual framework is to conduct extensive empirical studies using diverse datasets and real-world scenarios to validate the chosen machine-learning

algorithm. This involves testing the ML algorithm across various domains and data characteristics to ensure its robustness and reliability. Empirical validation is crucial to demonstrate the performance of the ML algorithm in real time. By providing empirical evidence for the performance of ML algorithms, the conceptual framework can build confidence among practitioners regarding the applicability of ML algorithms to real-world problems. Finally, in the last step, the conceptual framework intends to provide actionable insights and guidelines that help practitioners make better-informed decisions when selecting and deploying machine-learning algorithms for specific tasks. This involves offering recommendations based on a comprehensive efficiency analysis, highlighting the strengths and weaknesses of different algorithms in various contexts. By enhancing decision making, the conceptual framework aims to ensure that practitioners can choose the most appropriate algorithms for their needs, leading to more effective and efficient solutions to real-world problems.

The proposed conceptual framework for the efficiency analysis of machine-learning algorithms is based on several key principles designed to address the limitations of traditional evaluation methods. It emphasizes comprehensive evaluation metrics beyond accuracy to include computational efficiency, scalability, robustness, resource utilization, and ease of implementation. This holistic approach ensures a thorough assessment of multiple dimensions. The conceptual framework prioritizes simplification for accessibility, making the evaluation process user friendly for practitioners with varying levels of expertise. It offers clear guidelines and automated tools to facilitate rapid and effective comparisons without complex analyses. Empirical validation underpins the conceptual framework and involves extensive testing across diverse datasets and real-world scenarios. This empirical approach ensures the robustness and practical utility of the conceptual framework, accurately reflecting the performance of the algorithm in real applications. Additionally, the conceptual framework provides actionable insights for decision making, offering detailed comparative analysis and tailored recommendations based on specific needs, aiding practitioners in selecting the most suitable algorithms. Finally, the conceptual framework incorporates iterative refinement and feedback, allowing continuous updates based on user experience and new developments in machine learning. This ensures that the conceptual framework remains relevant and effective, adapting to evolving best practices and technological advancements.

## IMPLEMENTATION OF THE CONCEPTUAL FRAMEWORK

The efficiency of ML algorithms is critical for practical applications, particularly in environments with constrained resources or real-time requirements. Efficiency encompasses various factors such as computational speed, memory usage, and resource utilization, all of which affect the performance of the algorithm. To systematically evaluate and compare the efficiencies of different ML algorithms, we employ a composite efficiency score that integrates multiple key metrics. This approach ensures a comprehensive assessment, allowing informed decision-making in selecting the most suitable algorithm for a given task. The workflow of the proposed implementation process is illustrated in Fig. 2.

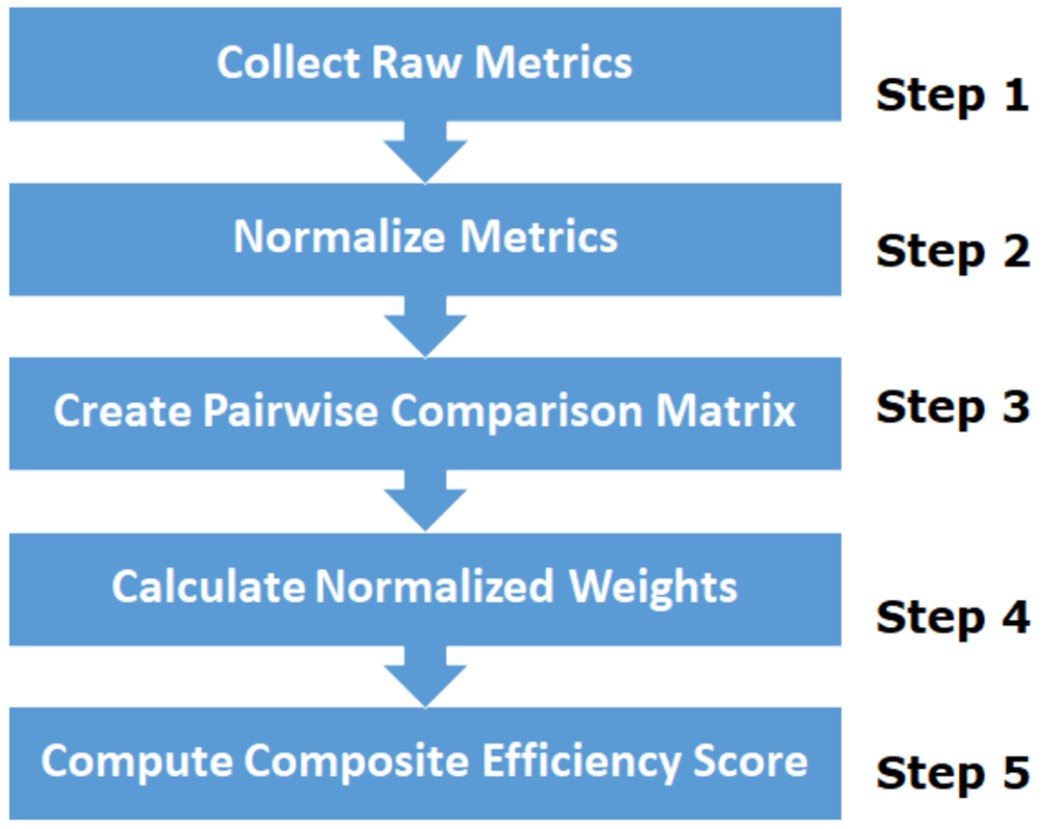

**Figure 2** **Working principle for efficiency analysis of ML algorithms.** Workflow of ML efficiency analysis.

    Figure 2 shows the workflow for evaluating the efficiency of the machine learning algorithms. In Step 1, "Collect Raw Metrics", we gather the raw values for key metrics such as training time, prediction time, memory usage, CPU usage, GPU usage, RAM usage, scalability, robustness, and adaptability. In Step 2, "Normalize Metrics", each metric is normalized to a scale of 0 to 1 using the min-max normalization formula to ensure comparability. In Step 3, "the Analytic Hierarchy Process (AHP)" is used to Create Pairwise Comparison Matrix that reflects the relative importance of each metric. In Step 4, "Calculate Normalized Weights", the pairwise comparison matrix is normalized and the average of each row is calculated to determine the weights. Finally, in Step 5, "Compute Composite Efficiency Score", the composite efficiency score formula is applied using the normalized metrics and their corresponding weights, resulting in a comprehensive efficiency assessment for the machine learning algorithms. The following sections detail the workflow illustrated in Fig. 2.

## Key metrics

We identified these key metrics based on the foundational study of *Zhou et al. (2017)*. Drawing from the insights and findings of this previous research, the authors carefully selected metrics, such as training time, prediction time, memory usage, CPU and GPU

utilization, scalability, robustness, and adaptability. Building on the comprehensive analysis conducted by *Zhou et al. (2017)* they ensured that the evaluation conceptual framework encompasses all critical aspects of algorithmic performance. The evaluation of the ML algorithm efficiency involves several key metrics, each representing a different aspect of performance. These metrics are denoted as follows.

- **Training time ($T_{\text{train}}$):** The time required to train the algorithm on a dataset.
- **Prediction time ($T_{\text{predict}}$):** The time required to make predictions using the trained model.
- **Memory usage ($M$):** The amount of memory consumed during training and prediction.
- **CPU usage ($U_{\text{CPU}}$):** Percentage of CPU resources utilized during algorithm execution.
- **GPU usage ($U_{\text{GPU}}$):** Percentage of GPU resources utilized, if applicable.
- **RAM usage ($U_{\text{RAM}}$):** The amount of RAM consumed during algorithm execution.
- **Scalability (S):** The algorithm's ability to handle increasing data sizes efficiently.
- **Robustness (R):** The algorithm's ability to maintain performance despite variations in the data quality or distribution.
- **Adaptability (A):** Ease with which the algorithm can be adapted to different datasets or requirements.

## Normalizing metric values

To integrate these metrics into a single efficiency score, we first normalized each metric to a scale of 0 to 1. Normalization ensures that different metrics, which may have varying units and ranges, can be meaningfully compared and combined.

The normalization formula for a metric $x$ is:

$$N(x) = \frac{x - x_{min}}{x_{max} - x_{min}} \tag{1}$$

where $x_{min}$ and $x_{max}$ are the minimum and maximum observed values for the metric $x$.

## Significance of weights

The weights in the composite efficiency formula reflect the relative importance of each metric in the overall assessment of the efficiency. Assigning appropriate weights is crucial, as it influences the final efficiency score and ensures that the evaluation aligns with the specific requirements and priorities of the application. For instance, in real-time systems, prediction time might be given more weight than training time, whereas in applications requiring adaptability, the ability to adjust to different datasets might be prioritized.

### Assigning weights using AHP

The Analytic Hierarchy Process (AHP) is a structured method for determining the weights of each metric (*Saaty, 2001*) based on their relative importance. AHP involves the following steps.

### Creating pairwise comparison matrix

In AHP, a pairwise comparison matrix is created to compare the importance of each metric relative to the others. As shown in Table 1, the AHP pairwise comparison matrix presents the relative importance of each metric for weight assignment. Each element $a_{ij}$ in the matrix

**Table 1  AHP pairwise comparison matrix for metric weight assignment.**

| | $T_{Train}$ | $T_{Predict}$ | $M$ | $U_{CPU}$ | $U_{GPU}$ | $U_{RAM}$ | $S$ | $R$ | $A$ |
|---|---|---|---|---|---|---|---|---|---|
| $T_{Train}$ | 1 | $a_{12}$ | $a_{13}$ | $a_{14}$ | $a_{15}$ | $a_{16}$ | $a_{17}$ | $a_{18}$ | $a_{19}$ |
| $T_{Predict}$ | $\frac{1}{a_{12}}$ | 1 | $a_{23}$ | $a_{24}$ | $a_{25}$ | $a_{26}$ | $a_{27}$ | $a_{28}$ | $a_{29}$ |
| $M$ | $\frac{1}{a_{13}}$ | $\frac{1}{a_{23}}$ | 1 | $a_{34}$ | $a_{35}$ | $a_{36}$ | $a_{37}$ | $a_{38}$ | $a_{39}$ |
| $U_{CPU}$ | $\frac{1}{a_{14}}$ | $\frac{1}{a_{24}}$ | $\frac{1}{a_{34}}$ | 1 | $a_{45}$ | $a_{46}$ | $a_{47}$ | $a_{48}$ | $a_{49}$ |
| $U_{GPU}$ | $\frac{1}{a_{15}}$ | $\frac{1}{a_{25}}$ | $\frac{1}{a_{35}}$ | $\frac{1}{a_{45}}$ | 1 | $a_{56}$ | $a_{57}$ | $a_{58}$ | $a_{59}$ |
| $U_{RAM}$ | $\frac{1}{a_{16}}$ | $\frac{1}{a_{26}}$ | $\frac{1}{a_{36}}$ | $\frac{1}{a_{46}}$ | $\frac{1}{a_{56}}$ | 1 | $a_{67}$ | $a_{68}$ | $a_{69}$ |
| $S$ | $\frac{1}{a_{17}}$ | $\frac{1}{a_{27}}$ | $\frac{1}{a_{37}}$ | $\frac{1}{a_{47}}$ | $\frac{1}{a_{57}}$ | $\frac{1}{a_{67}}$ | 1 | $a_{78}$ | $a_{79}$ |
| $R$ | $\frac{1}{a_{18}}$ | $\frac{1}{a_{28}}$ | $\frac{1}{a_{38}}$ | $\frac{1}{a_{48}}$ | $\frac{1}{a_{58}}$ | $\frac{1}{a_{68}}$ | $\frac{1}{a_{78}}$ | 1 | $a_{89}$ |
| $A$ | $\frac{1}{a_{19}}$ | $\frac{1}{a_{29}}$ | $\frac{1}{a_{39}}$ | $\frac{1}{a_{49}}$ | $\frac{1}{a_{59}}$ | $\frac{1}{a_{69}}$ | $\frac{1}{a_{79}}$ | $\frac{1}{a_{89}}$ | 1 |

represents the relative importance of metric i compared to metric j using a scale from 1 to 9. A scale ranging from one to nine was used to rate the relative importance of each pair of metrics. A rating of 1 signifies that the metrics are equally important, whereas a rating of 9 indicates that one metric is extremely important. Ratings of 3, 5, and 7 denote increasing degrees of importance, ranging from moderately to extremely important, respectively. This scale provides a structured conceptual framework for comparing the significance of different metrics and facilitating a nuanced assessment of their relative importance in the evaluation process.

## Calculating normalized weights

To calculate the normalized weights, follow these steps:

1. **Sum each column**: Calculate the sum of each column in the pairwise comparison matrix.
2. **Normalize the matrix**: Divide each element by the sum of its respective column.
3. **Calculate the average of each row** is calculated, and the average of the normalized values in each row gives the normalized weight for each metric.

## Composite efficiency score

The composite efficiency score, EEE, was calculated as the weighted sum of the normalized metrics:

$$
\begin{aligned}
E = &\, w_1 \cdot N\left(T_{Train}\right) + w_2 \cdot N\left(T_{Predict}\right) + w_3 \cdot N(M) + w_4 \cdot N\left(U_{CPU}\right) + w_5 \cdot N\left(U_{GPU}\right) \\
&+ w_6 \cdot N\left(U_{RAM}\right) + w_7 \cdot N(S) + w_8 \cdot N(R) + w_9 \cdot N(A)
\end{aligned}
\tag{2}
$$

where $w_i$ are the weights assigned to each normalized metric $N(x_i)$.

## Discussion on implementation

From Fig. 1, "Conceptual Framework for Efficiency Analysis of Machine Learning (ML) Algorithms", and Fig. 2, "Workflow for Efficiency Analysis of ML Algorithms", we have implemented the workflow on medical image data and agriculture crop prediction data, as detailed in Appendix A and Appendix B, respectively. The medical image data

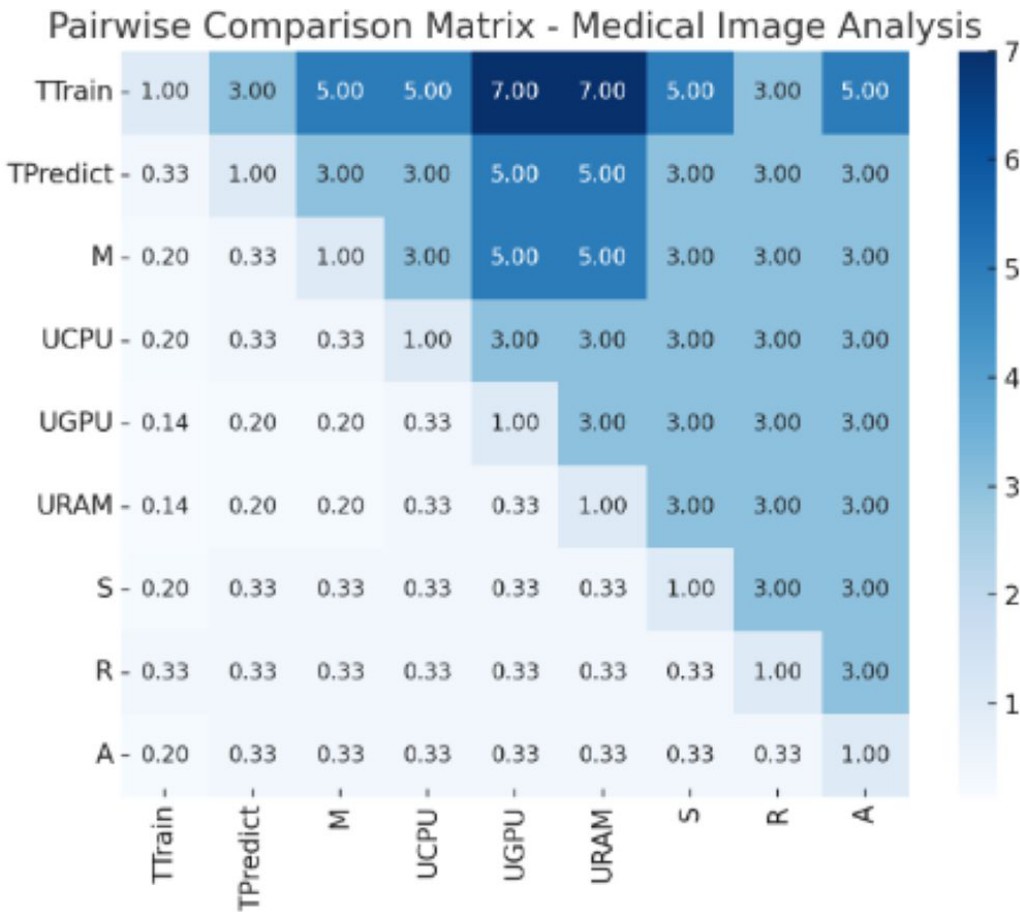

**Figure 3  Normalized metrics for medical image analysis.** Values of the normalized metrics for medical image analysis.

used in our experiments was the BraTS 2020 dataset, which focuses on brain tumor segmentation. This dataset includes multimodal MRI scans, such as T1, T1Gd, T2, and FLAIR, with labels highlighting tumor regions (enhancing tumor, tumor core, and edema), making it suitable for segmentation and survival prediction tasks. For each MRI scan, key features, such as the presence of different tumor regions, were considered in the prediction task. The crop prediction dataset, sourced from Kaggle, included environmental factors and soil composition data such as nitrogen (N), phosphorus (P), potassium (K), temperature, humidity, pH, and rainfall. This dataset enables machine-learning algorithms to recommend optimal crops based on these features, thereby improving agricultural productivity. Data visualizations of the normalized metrics, Pairwise comparison matrix and composite efficiency score for medical image data is presented in Figs. 3, 4 and 5 located in Appendix A. Similarly, the data visualizations of the normalized metrics, Pairwise comparison matrix and composite efficiency score for crop prediction data is presented in Figs. 6, 7 and 8 located in Appendix B.

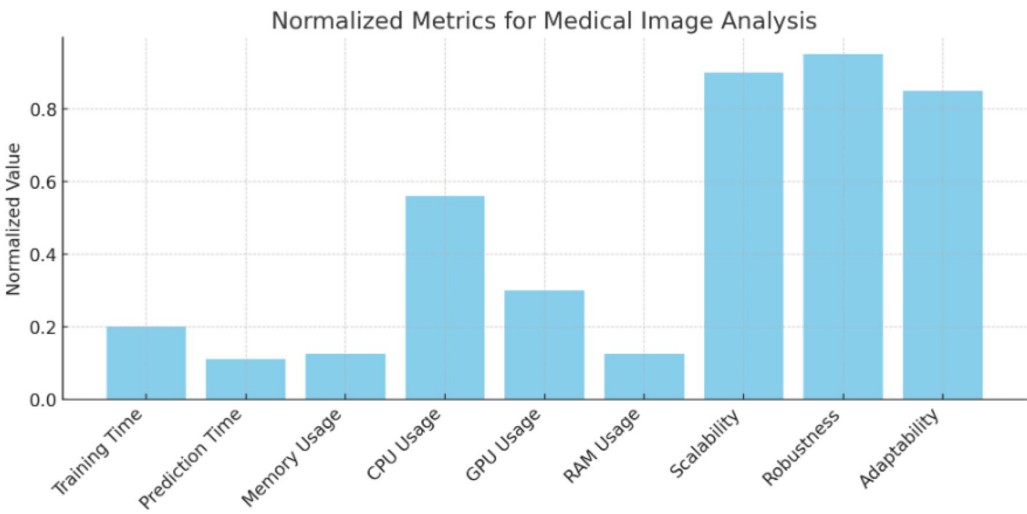

**Figure 4** **Pair wise comparison matrix for medical image analysis.** Values of pair wise comparison matrix for medical image analysis.

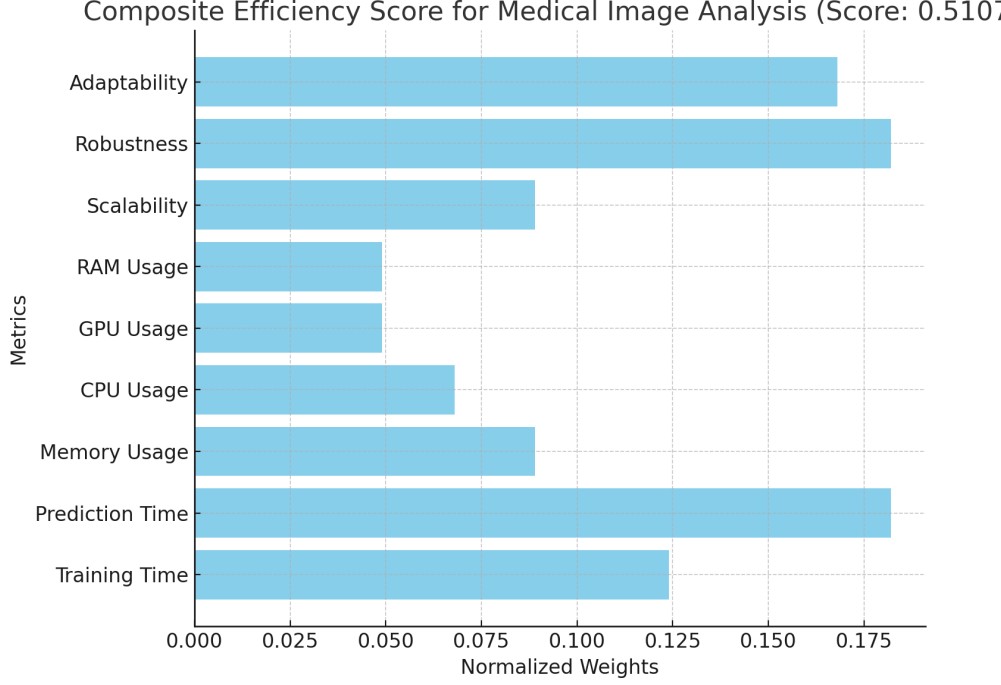

**Figure 5** **Composite efficiency score for medical image analysis.** Proposition of composite efficiency score for medical image analysis.

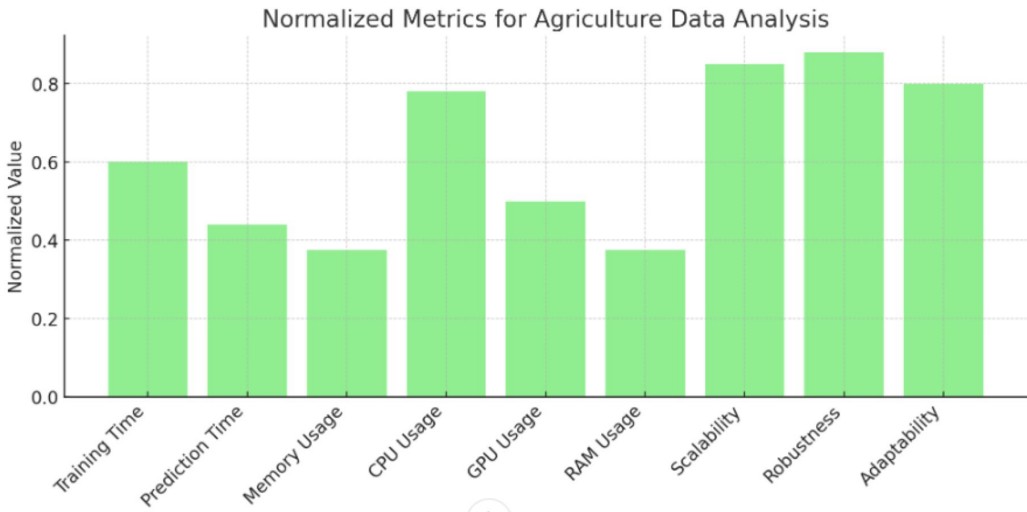

**Figure 6** **Normalized metrics for agriculture data analysis.** Values of normalized metrics for agriculture data analysis.

We employed a diverse set of machine-learning algorithms, including random forests, support vector machines (SVMs), and neural networks, chosen for their known efficiency in handling the characteristics of the two datasets. Random Forests, for instance, are effective in both structured and unstructured data and are known for their robustness against overfitting and interpretability, which makes them suitable for crop prediction datasets. In this context, random forests helps provide reliable crop recommendations based on varying environmental and soil conditions, as it can handle the multi-dimensionality of the dataset while maintaining strong generalization capabilities.

SVMs handle high-dimensional data with complex relationships between features, which are essential in medical image analysis. SVMs can effectively classify tumor regions based on MRI data, allowing for accurate segmentation and diagnosis by identifying the optimal hyperplane to differentiate between healthy and tumor regions. Their capability to manage nonlinearly separable data, which is often the case in medical diagnostics, makes them highly effective in this application.

Finally, neural networks were employed because of their versatility and ability to model complex nonlinear relationships between inputs and outputs. In both medical image analysis and crop prediction, neural networks can automatically learn relevant features from raw data, without requiring extensive manual feature engineering. In medical diagnostics, neural networks can capture subtle patterns in MRI images, which may be difficult to detect using traditional methods, thus improving the diagnostic accuracy. In agriculture, they allow for dynamic learning from varied environmental data, adapting to the complex interplay of factors, such as soil nutrients and climate.

The rationale behind selecting these algorithms is their ability to handle the unique challenges posed by both datasets. Random forests offers high interpretability and resilience to overfitting, making it suitable for structured datasets, such as crop prediction. SVMs

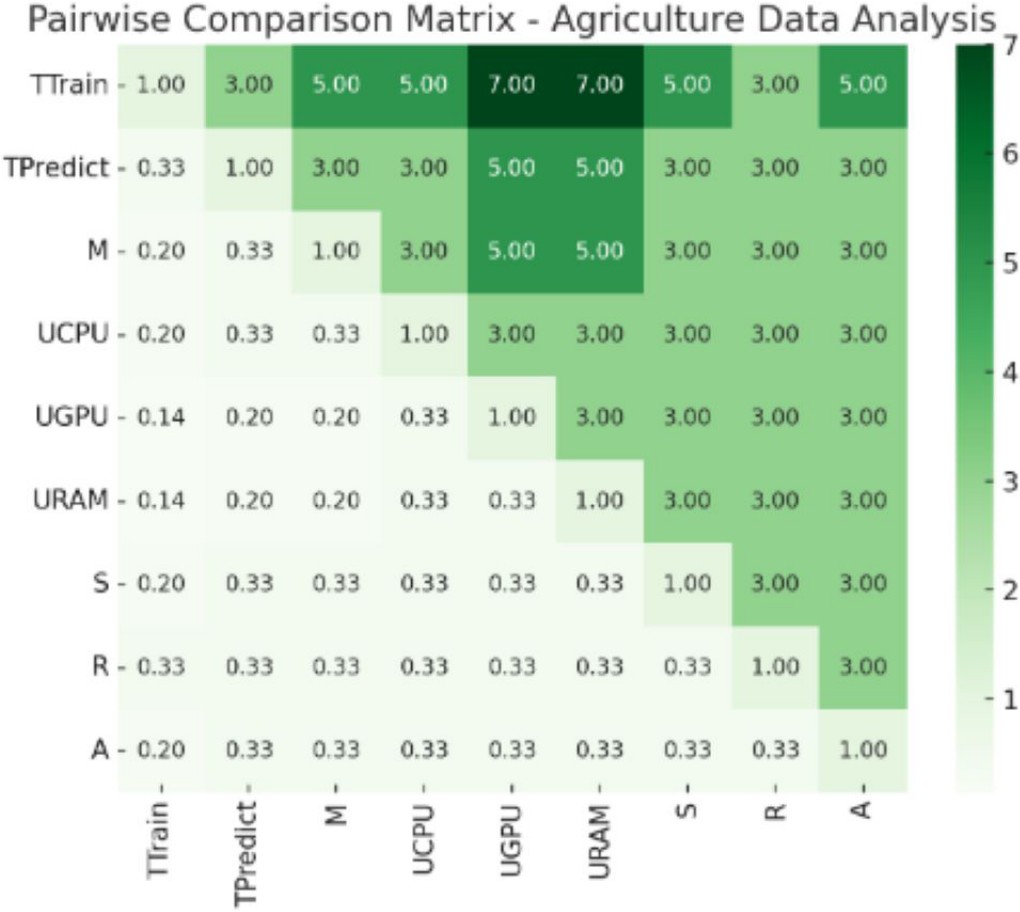

**Figure 7** **Pair wise comparison matrix—agriculture data analysis.** Values of pair wise comparison matrix—agriculture data analysis.

were chosen for their performance in handling high-dimensional and complex data, making them ideal for segmentation in medical image analysis. Neural networks, which are known for their ability to automatically extract features from complex datasets, provide the necessary adaptability for both applications. These algorithms ensure that the workflow aligns with the unique demands of each application, providing a comprehensive evaluation of the algorithmic performance.

The composite efficiency score calculated for each algorithm allowed us to assess their performance across different datasets, ensuring that the algorithm selection was aligned with both domain requirements and computational efficiency.

## CONTRIBUTIONS TO RESEARCH

This study offers a novel conceptual framework for analyzing the efficiency of machine learning algorithms and presents several key contributions that advance the field. One significant contribution lies in the conceptual framework's emphasis on identifying a set of key efficiency metrics. Traditionally, machine-learning evaluations often involve extensive

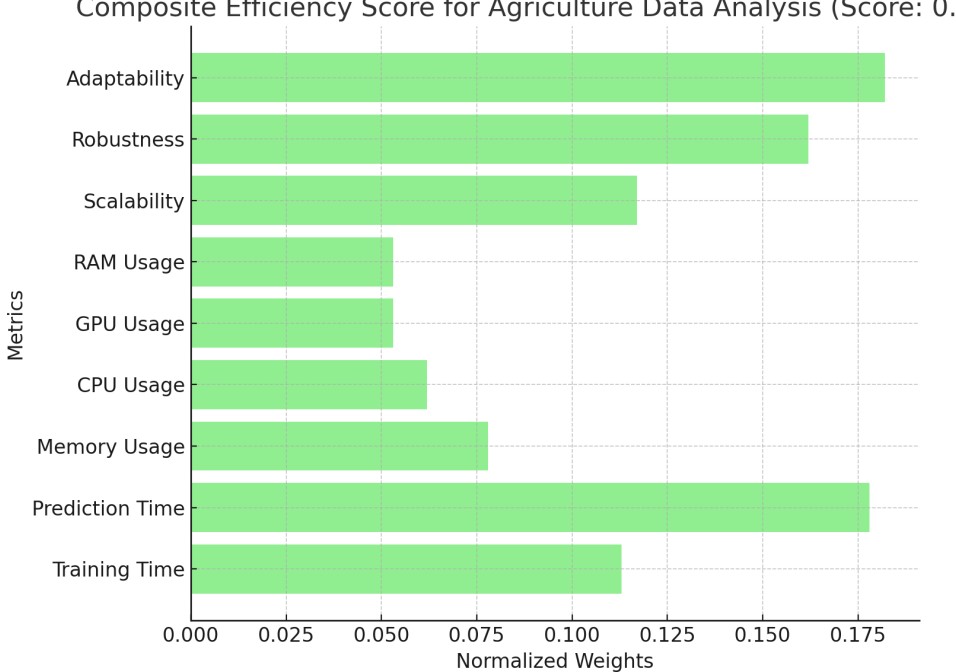

**Figure 8  Composite efficiency score of agriculture data analysis.** Proposition of composite efficiency score of agriculture data analysis.

analysis of numerous metrics. This approach is overwhelming and time consuming. Our conceptual framework proposes a shift towards identifying a smaller set of metrics that directly affect the performance of a specific application. By focusing on these critical metrics, we can streamline the evaluation process and obtain a clearer picture of how well an algorithm performs in the real world. This targeted approach allows practitioners to make informed decisions about which algorithm to choose and how to optimize for their specific needs.

Furthermore, this study breaks new ground by demonstrating the applicability of the conceptual framework in real-time scenarios. Many existing evaluation methods are static, providing a snapshot of performance at a single point in time. However, real-world machine-learning applications often operate in dynamic environments. Our conceptual framework addresses this issue by focusing on metrics that can be continuously measured as the system runs. This allows for ongoing evaluation and adjustments to the machine-learning workflow based on real-time data and user behavior. For instance, the conceptual framework can be used to monitor recommendation accuracy in a recommendation system or to track model latency in a fraud-detection system. This real-time feedback loop is crucial for ensuring that the chosen machine learning algorithm remains efficient and effective over time.

The conceptual framework also empowers practitioners to make improved decisions regarding machine-learning algorithms. By highlighting critical efficiency metrics, this approach aids in selecting the best algorithm for a specific application's needs and

resource constraints. For example, if a recommendation system prioritizes speed over perfect accuracy, the conceptual framework can help identify algorithms that deliver fast recommendations, even if they have a slightly lower accuracy score. This focus on application-specific efficiency metrics ensures that practitioners do not simply chase the highest possible accuracy on a benchmark dataset but rather select algorithms that are well-suited for real-world tasks at hand.

Finally, the simplicity and ease of implementation of this conceptual framework contributes to its potential for broader adoption. The conceptual framework is designed to be accessible to a wider range of practitioners, including those with limited machine learning expertise. By offering a clear and straightforward approach, the conceptual framework has the potential to broaden the adoption of efficient machine-learning practices across various disciplines. This wider adoption can lead to the development of more efficient and effective machine-learning solutions for a wider range of applications. Overall, this study offers a valuable contribution to the field of machine learning efficiency analysis by proposing a simplified, real-time applicable conceptual framework that empowers data scientists and developers to make data-driven decisions and optimize their machine learning workflows.

## Implications for practice

This study introduces a simplified conceptual framework for analyzing the efficiency of machine learning algorithms. This conceptual framework holds significant implications for practitioners across various fields, empowering them to make informed decisions and optimize their machine learning workflows. Below, let us delve deeper into these implications:

In practice, the evaluation of machine-learning algorithms remains a daunting task. This conceptual framework proposes a paradigm shift by identifying a focused set of key efficiency metrics. These metrics are directly related to the performance of specific applications, allowing practitioners to move beyond generic accuracy benchmarks. By prioritizing these critical metrics, practitioners can quickly assess different algorithms and make informed decisions about which one best aligns with their project's needs and resource constraints. This targeted approach significantly reduces the time and effort required for the algorithm selection, leading to faster development cycles.

A significant advantage of this conceptual framework is its emphasis on metrics that reflect the performance of an algorithm within a deployed system. Many traditional evaluation methods focus solely on the accuracy of benchmark datasets, which may not always translate to real-world effectiveness. This conceptual framework addresses this issue by prioritizing metrics such as training time, latency (response speed), and user interaction with recommendations (in the case of recommendation systems). This data-driven approach ensures that practitioners do not simply chase the highest possible accuracy on a static dataset but rather select algorithms that function efficiently and deliver value in the real world. For instance, a recommendation system may prioritize speed over perfect accuracy. The conceptual framework can help identify algorithms that deliver fast recommendations, even if they have a slightly lower accuracy score. This ensures that the users receive recommendations quickly and efficiently.

Machine learning efficiency is not only about accuracy; it also considers the resources available for training and deployment. This conceptual framework empowers practitioners to make informed decisions on resource allocation. By identifying algorithms that deliver the desired performance within the available resources, practitioners can optimize their workflow. For example, if a project has limited computational power, the conceptual framework can help to identify algorithms that achieve acceptable accuracy with shorter training times. This allows practitioners to allocate resources more effectively and to avoid bottlenecks caused by computationally expensive algorithms. Focusing on key metrics and real-time applicability allows practitioners to establish faster experimentation and iteration cycles. This streamlined approach enables practitioners to quickly evaluate different algorithms, identify the best option based on chosen metrics, and monitor their performance in real time as the system runs. The ability to gather continuous feedback through the conceptual framework allows for ongoing optimization and ensures that the chosen algorithm remains efficient as the data and user behavior evolve. This iterative approach is crucial for maintaining the effectiveness of machine-learning models in dynamic environments.

A key strength of this conceptual framework is its simplicity and ease of use. This makes it accessible to a wider range of practitioners, even those with limited machine learning expertise. Previously, complex evaluation methods may have created barriers to entry for some teams. This conceptual framework empowers them to actively participate in the selection and optimization process, bringing valuable perspectives from different areas of expertise. Additionally, the conceptual framework can be easily integrated into existing workflows to minimize disruption and promote the faster adoption of efficient machine learning practices across various disciplines. Thus, the proposed conceptual framework enables practitioners to navigate the complexities of machine learning algorithm selection and optimization. By focusing on key efficiency metrics and real-time performance, practitioners can make data-driven decisions that deliver real-world value. This approach not only streamlines the workflow but also allows for optimal resource allocation and continuous improvement through faster experimentation cycles. The conceptual framework's accessibility further broadens its reach, fostering the adoption of efficient machine-learning practices across a wider range of fields.

## Validity of the conceptual framework

The validity of the proposed conceptual framework for the efficiency analysis of ML algorithms is established through several key factors. The conceptual framework employs comprehensive evaluation metrics, extending beyond accuracy to include computational efficiency, scalability, robustness, resource utilization, and ease of implementation, thus ensuring a holistic assessment. Empirical validation through extensive testing across diverse datasets and real-world scenarios provides a conceptual framework for practical performance data, thereby enhancing its reliability and applicability. Prioritizing simplification and accessibility, the conceptual framework provides clear guidelines, thereby making it user-friendly for practitioners with varying expertise levels and ensuring a consistent application. It also delivers actionable insights through detailed comparative

analysis and tailored recommendations, aiding informed decision making and the selection of suitable algorithms for specific applications. By incorporating mechanisms for iterative refinement and feedback, the conceptual framework remains current, with evolving best practices and technological advancements, maintaining its relevance and effectiveness. This continuous adaptation ensures that the conceptual framework provides accurate and reliable evaluation over time. The combination of empirical validation, comprehensive metrics, user-friendly design, actionable insights, and iterative refinement ensures the robustness and practical utility of the conceptual framework. These elements collectively support the ability of the conceptual framework to offer reliable, accurate, and practical assessments of ML algorithms, making it an essential tool for practitioners and researchers.

## CONCLUSIONS AND FUTURE RESEARCH

This study presents a simplified conceptual framework for analyzing the efficiency of machine learning algorithms, aimed at providing a streamlined and practical approach for evaluating model performance. The framework emphasizes the identification of key efficiency metrics that directly impact the performance of specific applications, thus enabling practitioners to make more informed decisions when selecting and optimizing machine learning models. By concentrating on a curated set of relevant metrics, this approach simplifies the evaluation process, ensuring that it is both manageable and meaningful.

Our demonstration of the applicability of the framework through a real-world example of a recommendation system highlights its practical benefits. The framework facilitates the real-time monitoring of critical metrics, such as training time, latency, and user interaction with recommendations. This capability allows for continuous evaluation and data-driven adjustments, ensuring that the machine-learning workflow remains adaptive and responsive to evolving data and user behaviors.

The advantages of this conceptual framework are notable; its simplicity and ease of implementation make it accessible to a broad audience of practitioners, from novices to experts. This encourages a focus on real-world performance metrics that are crucial for the success of machine learning applications. Moreover, the real-time analysis component supports ongoing improvement and adaptation, enhancing the system's ability to meet changing demands and optimizing user experiences.

While this research focuses on a simplified approach, it sets the stage for future investigation. Future research could explore the integration of more complex efficiency metrics to enrich the analysis or examine how the framework can be applied to a broader range of machine learning tasks beyond recommendation systems. Additionally, addressing ethical considerations such as data privacy, bias mitigation, and regulatory compliance is essential to ensure that machine learning technologies are used responsibly and equitably. By incorporating these elements, future work will contribute to a more comprehensive and ethically grounded approach to machine learning efficiency, further advancing the field and promoting responsible innovation.

### Funding

The authors received no funding for this work.

### Competing Interests

The authors declare there are no competing interests.

### Author Contributions

- Muthuramalingam Sivakumar conceived and designed the experiments, analyzed the data, authored or reviewed drafts of the article, and approved the final draft.
- Sudhaman Parthasarathy conceived and designed the experiments, performed the experiments, analyzed the data, performed the computation work, prepared figures and/or tables, authored or reviewed drafts of the article, and approved the final draft.
- Thiyagarajan Padmapriya performed the experiments, analyzed the data, performed the computation work, prepared figures and/or tables, and approved the final draft.

### Data Availability

The code is available in the Supplemental File.

The data is available at:

- RSNA-ASNR-MICCAI Brain Tumor Segmentation (BraTS) Challenge 2021: http://braintumorsegmentation.org/
- https://github.com/Chandradithya8/FarmApp.

### Supplemental Information

Supplemental information for this article can be found online at http://dx.doi.org/10.7717/peerj-cs.2418#supplemental-information.

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
