# Peer review of "A simplified approach for efficiency analysis of machine learning algorithms"

_PeerJ Computer Science, doi:10.7717/peerj-cs.2418_

## Round 0.1 · original submission · Major Revisions

Dear authors,

Your manuscript has not been accepted in its current form. It is expected that the necessary edits and additions will be made according to the comments of the reviewers. Reviewer 1 has asked you to provide specific reference. You are welcome to add it if you think it is useful and relevant. However, you are under no obligation to include it, and if you do not, it will not affect my decision.

Best wishes,

Reviewer 1 ·

Basic reporting

Clarity and Professionalism of Language:

Overall, the language used is clear and professional. However, there are instances where the text could benefit from minor grammatical improvements to enhance readability. For example, phrases such as "By prioritizing these critical metrics" could be made more concise.
Literature References and Context:

The article provides a good background and references relevant literature. However, the authors could include more recent studies to ensure the context is up-to-date. Specifically, including references from the past two years could strengthen the literature review.
Article Structure and Figures:

The structure of the article is logical and conforms to academic standards. However, some figures (like Figure 1 and Figure 2) could be accompanied by more detailed captions explaining their relevance and content in depth.
Self-Contained and Relevant Results:

The article is self-contained and presents results relevant to the hypotheses. Yet, the introduction could be expanded to better define key terms and concepts used throughout the paper, ensuring that all readers have a clear understanding from the start.

Experimental design

Originality and Scope:

The research fits well within the journal's aims and scope. The research question is relevant and meaningful, addressing a significant gap in the existing literature regarding the efficiency analysis of machine learning algorithms.
Technical and Ethical Standards:

The investigation appears to be rigorous and conducted to high technical standards. However, it would be beneficial to provide more details on any ethical considerations, especially concerning data use and privacy if applicable.
Methodological Detail and Replicability:

The methods are described in detail, but some sections could benefit from further elaboration. For example, more comprehensive step-by-step instructions or additional supplementary material could help other researchers replicate the study more easily.

Validity of the findings

Validity of the Findings
Data Robustness and Statistical Soundness:

The data provided are robust and the statistical methods sound. Nonetheless, the authors should consider adding more statistical tests to validate their findings further. Including confidence intervals or p-values where relevant could strengthen the statistical analysis.
Clear and Linked Conclusions:

The conclusions are generally well stated and linked to the research question. However, the authors could improve the discussion section by exploring the broader implications of their findings in more detail and suggesting future research directions.

Additional comments

Specific Suggestions for Improvement
Improve Language and Clarity:

Review and revise the manuscript for minor grammatical errors and improve the clarity of complex sentences. Consider using a professional editing service for polishing.
Update Literature Review:

Add more recent references from the last two years to ensure the literature review is current and comprehensive.
Enhance Figure Captions and Explanations:

Provide more detailed captions for figures and explain their relevance more thoroughly within the text.
Expand Introduction and Define Terms:

Expand the introduction to include clear definitions of all key terms and concepts. This will make the paper more accessible to a broader audience.
Detail Ethical Considerations:

Include more information on ethical considerations related to data use and privacy, ensuring that the study adheres to ethical standards.
Additional Statistical Validation:

Incorporate additional statistical tests to further validate the findings. Include confidence intervals or p-values where appropriate.
Broaden Discussion and Future Research:

Expand the discussion to cover the broader implications of the findings and suggest directions for future research.
By addressing these points, the authors can significantly enhance the quality and impact of their manuscript.

In the introduction, I would add a look from the medical side at AI. Please read the article and quote from it: DOI10.3390/diagnostics13152582
Also missing is the information that the biggest leap in AI development in the medical field was made during the Covid-19 pandemic.

Cite this review as

Reviewer 2 ·

Basic reporting

1. Most of the citations are over old research which poses a concerns of whether the research questions considered in this study are still valid or unanswered in the literature already.

2. Some key statement made in the introductory section are clearly NOT the authors(s) own words but were not appropriately cited.

3. A research article should have ONLY one goal(aim) and then many objectives that can help fulfil the goal or aim. Pg no 139 - 141.

4. Firstly, the current related works section lacks a clear definition and needs substantial revision before this paper can be considered for publication. The authors' disproportionate focus on 'time complexity,' which was only briefly mentioned in the previous section, suggests a potential bias over other crucial metrics. Secondly, the authors' exclusive reliance on AutoML, disregarding other tools and services like Amazon SageMaker Canvas that perform similar functions, raises concerns about their understanding of the field. Critically, a comprehensive related works section should review existing literature in the research area, detailing the methodologies, limitations, and contributions of each study, and provide a thoughtful evaluation of how the authors' approach differs or improves upon these studies. This approach is essential and must be implemented.

Experimental design

1. It is important to note that this is a conceptual framework. Therefore, the term 'framework' in this study should be consistently referred to as a conceptual framework.

2. The study lacks any justification for the selection of the methods employed over other equally suitable alternatives.

Validity of the findings

The results are unconvincing and do not align with the discussions and implications presented. Firstly, there is a complete lack of information regarding the data used, and the reasons for this omission are unclear. It is also not specified which features (labels) were considered for each training and prediction. Furthermore, the paper fails to detail which machine learning algorithms were employed and the rationale behind their selection. No data preprocessing techniques were implemented, nor was there any indication of how top features were selected. Given these numerous issues, the outcomes cannot be considered objectively accurate, and thus the findings lack validation.

Cite this review as

Reviewer 3 ·

Basic reporting

The given manuscript is explaining a method for optimizing ML selection based on the consumption of hardware and train/test time based on the need of the application. They implemented a weighted matrix to combine various parameters to find an optimal solution for developing ML model adapted to hardware.
The work has minor scientific contributions however the methodology could be interesting for some people dealing with limited hardware capacities for train/test ML models.
Title should be more sophisticated showing some indication about the methodology. be more creative.

Abstract is just another introduction and not telling anything about method and results
Keywords are not selected properly

Experimental design

section 5.3 Significance of Weights seems to be made up. please bring references and show some robust ways of deciding about it.
most of the numerical examples are in the Appendix. maybe you could bring some figures with examples you have done rather than just a matrix. Be more creative. Too few figures are there.

The representation of matrix could be improved as well. the are not aligned sometimes.
mention several advantages in bullet form in conclusion.

Validity of the findings

The last paragraph of introduction and 1.1Motivation repeat same thing over and over again, and then all of them again repeated in section 2.Related works and 3.Research Objectives.
Discuss more results with figures. deeper analysis is required.

Cite this review as

---

## Round 0.2 · accepted · Accept

Dear authors,

Two of the original reviewers did not respond to the invitation for reviewing your revised manuscript. The other reviewer thinks your paper can be accepted. I also think that the paper has been sufficiently improved. As such, the article is considered acceptable.

Best wishes,

Reviewer 3 ·

Basic reporting

Good improvement.

Experimental design

Good improvement.

Validity of the findings

Good improvement.

Additional comments

Good improvement.

Cite this review as